# Episiotomy practices and associated factors in central Vietnam

**Hoang Thi Nam Giang**[1]*, **Do Thi Thuy Duy**[1], **Thanh-Huyen T. Vu**[2]*

**1** The University of Danang, School of Medicine and Pharmacy, Danang, Vietnam, **2** Department of Preventative Medicine, Northwestern University Feinberg School of Medicine, Chicago, Illinois

* huyenvu@northwestern.edu (THTV); htngiang@ud.edu.vn (HTNG)

## Abstract

### Introduction

Research on episiotomy practices in Vietnam is limited. This study aimed to describe episiotomy use and identify factors associated with its practice among vaginal births in Central Vietnam, following the implementation of restrictive episiotomy guidelines.

### Methods

We used data from a hospital-based, retrospective study conducted at Danang Hospital for Women and Children from April 2015 to March 2016. The study included all singleton, full-term vaginal births. Multivariable logistic regression was used to estimate the odds of episiotomy based on selected neonatal or maternal factors.

### Results

Among 3,471 eligible singleton births, 2,770 mothers (79.8%, 95% CI: 78.4% − 81.1%) underwent an episiotomy. The episiotomy rate was significantly higher in first-time births (97.7%, 95% CI: 96.8% − 98.3%) compared to second or subsequent births (61.5%, 95% CI: 59.2% − 63.8%), p<0.001. Multivariable analyses showed that first-time births, higher birth weight, younger maternal age, a less physical active occupation, and a history of miscarriage were significantly associated with higher odds of episiotomy. For example, the odds of episiotomy in first-time births was 24.21 (95% confidence interval (CI): 17.13 – 34.22) times higher than in second or subsequent births, and the odds for mothers with a history of miscarriage was 1.34 (95% CI: 1.03 – 1.73) compared to those without. Stratified analysis showed that these associations persisted in multiparous women but were not observed in primiparous women.

### Conclusion

This study highlights a very high episiotomy rate in Central Vietnam, especially among primiparous women despite recommendations from the World Health

**Data availability statement:** All data are available from the figshare database (accession number https://doi.org/10.6084/m9.figshare.27913800).

**Funding:** The author(s) received no specific funding for this work.

**Competing interests:** The authors have declared that no competing interests exist.

Organization and the Vietnam Ministry of Health. Several modifiable maternal and neonatal factors are linked to episiotomy use in multiparous women. While one year may not be sufficient to fully assess the restrictive policy's effect, the results of our study also suggest that the policy has not had a significant impact on reducing the episiotomy rate after one year. This indicates that, in addition to the policy, further support and interventions are needed to reduce episiotomy rate.

## Introduction

Episiotomy is a surgical incision with a scissor or a scalpel blade during last part of the second period of labor, and thereafter suture is required for correction [1]. During childbirth, vaginal tear may occur spontaneously, or a healthcare provider may perform an episiotomy to enlarge the vaginal outlet and facilitate the delivery [2]. Episiotomy is originally performed to prevent severe perineal tears. Routine episiotomy was first advocated by Pomeroy in 1918 [3] and became a common practice in obstetrics, despite a lack of clearly evidence supporting its effectiveness. Some suggested beneficial effects of episiotomy on mothers include a decrease in the possibility of third-degree vaginal tear [4], a reduction in the likelihood of fecal and urinary incontinence, and an improvement in sexual function [5]. Besides its maternal benefits, episiotomy is also suggested to reduce potential neonatal complications from a prolonged second stage of labor, such as fetal asphyxia, cranial trauma, and cerebral haemorrhage [2]. In cases of shoulder dystocia, episiotomy may provide more space to rotate the fetal shoulders. However, since 1970s, the necessity of routine episiotomy has been questioned [6]. Studies have shown that a restrictive approach to episiotomy is associated with similar or fewer complications compared to routine use [2], leading the World Health Organization (WHO) to recommend against routine episiotomy during spontaneous vaginal birth [7].

In Vietnam, the rate of episiotomy use and its associated factors are rarely reported despite the 2014 policy change that no longer recommends routine episiotomy [8]. Prior to 2014, only one survey conducted between 2012−13 in a maternity hospital in southern Vietnam to assess clinicians' practices regarding episiotomy [9]. In this study, among nulliparous women, the majority of obstetricians (82.6%) and midwives (98.7%) reported using episiotomies more than 90% of the time. Among multiparous women, a quarter of obstetricians reported using episiotomies under 60% of the time, while this rate for midwives was 3.8%. In 2014, the Vietnam Ministry of Health approved a technical guide on essential care for mothers and newborns during and immediately after birth (EENC), the practice of routine episiotomy is no longer recommended [8]. Since then, there is lacking data to evaluate the effect of this restrictive recommendation. In 2020, the rate of episiotomy was reported in a study conducting that focusing on breastfeeding in both southern and northern Vietnam, where 83.3% of mothers self-reported having undergone a vaginal birth with an episiotomy [10].

To our knowledge, no comprehensive study has been conducted on episiotomy practices and their associated factors in Vietnam, particularly in Central Vietnam. Moreover, although routine episiotomy has not been recommended since the implementation of EENC guideline in 2014, and EENC was introduced in Da Nang Hospital for Women and Children the same year, data on episiotomy practice during the early stage of this transition remain limited. Therefore, this study aimed to describe the prevalence of episiotomy among women who had vaginal births of single, full-term infants after one-year period of the implementation of restrictive episiotomy guidelines in Vietnam, and to identify the maternal and infant factors associated with this practice.

## Methods

### Study design

This is a hospital-based, cross-sectional study that collected data retrospectively.

### Study settings

Data were obtained from Danang Hospital for Women and Children (DHWC), one of the largest maternity hospitals in Central Vietnam, which serves as a referral center for a population of approximately 4 million [11].

### Data collection

We used a data collection form to extract information from hospital birth register books. The dataset included maternal characteristics such as year of birth, occupation, obstetric history, episiotomy practice (yes or no), as well as infant's sex and birth weight [12]. Data was accessed from April 2015 to March 2016 and the authors had no access to information that could identify individual participants during or after data collection.

### Inclusion criteria

The study included vaginal births that occurred at DHWC between April 2015 and March 2016. Only singleton, full-term births (≥37 weeks' gestation) were eligible for inclusion. Of the 3,491 vaginal, singleton, and full-term births, 20 were excluded due to missing values on variables of interest, resulting in a final analysis sample of 3,471 births. Since all births included in the study are singleton, the terms 'births' and 'mothers' are used interchangeably as units of analysis.

### Dependent and independent variables

The dependent variable in this study was episiotomy (yes vs. no). The independent variables included maternal age, maternal occupation, history of miscarriage (yes vs. no), parity (first-time birth vs. second or subsequent births), neonatal birth weight (in grams), and infant sex (male vs. female). Maternal occupation was categorized into three groups based on relative physical activity levels: high, moderate, and low. The high physical activity group included occupations such as soldier, security personnel, policewoman, postal worker, anesthesia, babysitter, chef, doctor, midwife, nurse, technician, tour guide, and farmer. The moderate physical activity group included garment workers and women employed on production lines for handmade goods. The low physical activity group comprised occupations such as accountant, auditor, banker, and clerk. Neonatal birth weight was categorized into quartiles: Quartile 1 (Q1): <2900 grams, Q2: 2900 grams to <3100 grams, Q3: 3100 grams to 3400 grams, and Q4: ≥3400 grams.

### Statistical analysis

Frequencies and percentages were used to describe the categorical variables. Means and 95% confidence interval (CI) or median and interquartile range (IQR) were used to describe continuous variables. T-tests or Chi-square tests were employed for overall group comparisons.

Logistic regression analyses were conducted to examine the association between maternal and infant characteristics and episiotomy practice. Binary logistic regression analysis was used to estimate the crude odds ratio (OR) and 95% CI between the episiotomy practice and maternal age, maternal occupation, history of miscarriage, parity, birth weight sex of the infant. Multivariable logistic regression was performed to identify factors associated with the episiotomy using adjusted OR (aOR) and 95% CI. Initially, logistic regression analyses were conducted for the entire sample. A stratified analysis was then performed based on parity (primipara and multipara). A P-value of <0.05 was considered statistically significant. All data analyses were carried out using R version 4.3.1 (R Foundation for Statistical Computing, Vienna, Austria)

### Ethical consideration

Ethical approval for the project was granted by the Scientific and Ethics Board of Danang Hospital for Women and Children and by the Ethics Committee of the Ludwig Maximilian University [12]. All data were fully anonymized, and the committee waived the need of informed consent from the patient.

### Results

Table 1 presents the maternal and neonatal characteristics in the analysis sample, both overall and by maternal parity at birth. Among the 3,471 births, the mean maternal age was 28.4, 2,770 (79.8%, 95% CI: 78.4% − 81.1%) involved an episiotomy. The episiotomy rate was 97.7% for first-time births (95% CI: 96.8% − 98.3%) and 61.5% for second or subsequent births (95% CI: 59.2% − 63.8%). All maternal and neonatal characteristics, except for infant sex, showed statistically significant differences between first-time births and second or subsequent births. For instance, younger mothers, those in low-physical activity occupations, mothers without a history of miscarriage, mothers of infants with lower birth weights, and those who underwent episiotomy tended to be first-time mothers.

Table 2 shows the results of the logistic regression analysis for factors associated with episiotomy in the entire sample. Multivariate analyses indicated that maternal age, occupation, history of miscarriage, parity, and neonatal birth weight were significantly associated with the likelihood of episiotomy. For example, mothers in low-physical activity occupation had 1.32 times the odds of undergoing an episiotomy compared to those in high-physical activity occupations (aOR: 1.32; 95% CI: 1.06–1.65). Mothers of neonates in the fourth quartile of birth weights (≥3400 grams) had an increased likelihood of episiotomy (aOR: 1.55; 95% CI: 1.17–2.05) compared to those in the first quartile. Notably, mothers of first-time births had significant higher odds of undergoing an episiotomy compared to those of second or subsequent births (aOR: 24.21, 95% CI: 17.13–34.22).

After stratifying by parity, none of the factors showed a significant association with episiotomy in the first-time birth group. In contrast, similar association were observed in the second or subsequent birth group, consistent with the findings in the entire analysis sample (Table 3).

### Discussion

This study included 3,471 singleton, full-term vaginal births in Central Vietnam during 2015–2016, one year following the implementation of restrictive episiotomy guidelines launched in the country. We observed a very high episiotomy rate, particularly among primiparous women. Factors such as neonatal birth weight, maternal age, occupation, and miscarriage history were associated with episiotomy among multiparous women, but not among primiparous women.

The high episiotomy rate in our study aligns with the finding from existing literature in other regions of the country. A survey conducted in 2012–2013 among obstetricians and midwives in southern Vietnam – prior to the implementation of restrictive episiotomy guidelines by the Vietnam Ministry of Health in 2014 – reported that 82.6% of obstetricians and 98.7% midwives performed episiotomies more than 90% of the time for first-time births [9]. Recent data also show that 83.3% of mothers in southern and northern Vietnam self-reported having undergone a vaginal birth with an episiotomy [10]. Our rate is also comparable to rates in other developing countries in the same region such as the Philippines (92%)

**Table 1. Maternal and neonatal characteristics, overall and by parity, n = 3,471.**

| Characteristics | All(n = 3,471) | First-time births (n = 1,756) | Second or subsequent births (n = 1,715) | P-value* |
|---|---|---|---|---|
| **Maternal characteristics** | | | | |
| **Age** (year) | | | | |
| Mean (95% CI) | 28.4 (28.2–28.5) | 25.9 (25.7–26.0) | 30.9 (30.7–31.1) | <0.001 |
| **Occupation,** n (%, 95% CI) | | | | |
| High physical activity | 1490 (42.9, 41.1–44.6) | 735 (41.9, 39.5–44.2) | 755 (44.0, 41.7–46.4) | 0.031 |
| Moderate physical activity | 788 (22.7, 21.3–24.1) | 381 (21.7, 19.8–23.7) | 407 (23.7, 21.7–25.8) | |
| Low physical activity | 1193 (34.4, 32.8–36.0) | 640 (36.4, 34.2–38.7) | 553 (32.3, 30.0–34.5) | |
| **History of miscarriage,** n (%, 95% CI) | | | | |
| No | 2994 (86.3, 85.1–87.4) | 1601 (91.2, 89.7–92.5) | 1393 (81.2, 79.3–83.0) | <0.001 |
| Yes | 477 (13.7, 12.6–14.9) | 155 (8.8, 7.5–10.3) | 322 (18.8, 17.0–20.7) | |
| **Neonatal characteristics** | | | | |
| **Neonatal birth weight** (grams) | | | | |
| Mean (95% CI) | 3128.9 (3116.8–3141.0) | 3052.5 (3036.8–3068.2) | 3207.1 (3189.3–3224.9) | <0.001 |
| **Catergory**, n (%, 95% CI) | | | | |
| Quartile 1: < 2900 | 767 (22.1, 20.7–23.5) | 484 (27.6, 25.5–29.7) | 283 (16.5, 14.8–18.3) | <0.001 |
| Quartile 2: 2900 to <3100 | 701 (20.2, 18.9–21.6) | 418 (23.8, 21.8–25.9) | 283 (16.5, 14.8–18.3) | |
| Quartile 3:3100 to <3400 | 1100 (31.7, 30.1–33.3) | 540 (30.7, 28.6–33.0) | 560 (32.7, 30.4–34.9) | |
| Quartile 4: ≥ 3400 | 903 (26.0, 24.6–27.5) | 314 (17.9, 16.1–19.8) | 589 (34.3, 32.1–36.6) | |
| **Neonatal sex**, n (%, 95% CI) | | | | |
| Male | 1755 (50.6, 48.9–52.2) | 882 (50.2, 47.9–52.6) | 873 (50.9, 48.5–53.3) | 0.690 |
| Female | 1716 (49.4, 47.8–51.1) | 874 (49.8, 47.4–52.1) | 842 (49.1, 46.7–51.5) | |
| **Episiotomy rate**, n (%, 95% CI) | | | | |
| No | 701 (20.2, 18.9–21.6) | 41 (2.3, 1.7–3.2) | 660 (38.5, 36.2–40.8) | <0.001 |
| Yes | 2770 (79.8, 78.4–81.1) | 1715 (97.7, 96.8–98.3) | 1055 (61.5, 59.2–63.8) | |

*P-values for overall group comparisons based on T-tests or Chi-square tests; CI: confidence interval.

and Cambodia (94.5%) [13,14]. While the WHO recommended restrictive episiotomy approach in uncomplicated deliveries, targeting a rate of 10% or lower [7], the episiotomy rate in our study exceeds this recommendation by more than eightfold.

It should be noted that Danang Hospital for Women and Children is one of the largest maternity hospitals in Central Vietnam, staffed by highly trained healthcare professional. The hospital was also among the first to implement the government's guidelines on restrictive episiotomy. However, one year after the implementation, the rate of episiotomy for first-time births remained nearly universal. This high rate maybe partly due to the perception that Vietnamese women have a shorter perineal length, potentially increasing their risk of perineal tears during labour [15], leading healthcare providers to maintain routine episiotomy practice. However, multiple studies have found no significant link between perineal length and the likelihood of needing an episiotomy [16,17]. Additionally, Vietnamese-born women giving birth in Australia experience significantly lower episiotomies compared to their counterparts in Vietnam, without an associated in adverse outcomes [18]. These findings suggest that the episiotomy rate in Vietnam could likely be reduced without compromising maternal or neonatal safety.

Our findings also suggest that additional underlying factors may be driving the continued high episiotomy rates. While first-time birth is a known predictor of episiotomy [19], the odds in our study was exceptionally high, as nearly all first-time

**Table 2. Adjusted odds ratios and 95% confidence intervals of episiotomy, n = 3,471.**

| Characteristics | aOR (95% CI) | P-value |
|---|---|---|
| **Age** (year) | 0.96 (0.94 - 0.98) | <0.001 |
| **Occupation Category** | | |
| Low-physical activity | 1.32 (1.06 - 1.65) | 0.011 |
| Moderate-physical activity | 1.13 (0.89 - 1.44) | 0.297 |
| High-physical activity | Reference | |
| **History of Miscarriage** | | |
| Yes | 1.34 (1.03 - 1.73) | 0.023 |
| No | Reference | |
| **Parity** | | |
| First-time birth | 24.21 (17.13 - 34.22) | <0.001 |
| ≥ Second-time birth | Reference | |
| **Neonatal birth weight (grams)** | | |
| Quartile 1: < 2900 | Reference | |
| Quartile 2: 2900 to <3100 | 0.97 (0.71 - 1.32) | 0.857 |
| Quartile 3: 3100 to <3400 | 1.19 (0.90 - 1.57) | 0.205 |
| Quartile 4: ≥ 3400 | 1.55 (1.17 - 2.05) | 0.002 |
| **Sex of infants** | | |
| Female | 0.99 (0.82 - 1.19) | 0.924 |
| Male | Reference | |

aOR: Adjusted Odds Ratio; CI: Confidence Interval.

**Table 3. Odds ratios and 95% confidence interval of episiotomy stratified by parity, n = 3,471.**

| Characteristics | First-time birth (n = 1,756) | | ≥Second-time birth (n = 1,715) | |
|---|---|---|---|---|
| | aOR (95% CI) | P-value | aOR (95% CI) | P-value |
| **Age** (year) | 0.92 (0.84 - 1.02) | 0.133 | 0.96 (0.94 - 0.98) | 0.003 |
| **Occupation Category** | | | | |
| Low physical activity | 0.60 (0.28 - 1.28) | 0.191 | 1.44 (1.14 - 1.81) | 0.001 |
| Moderate physical activity | 0.61 (0.26 - 1.43) | 0.264 | 1.20 (0.93 - 1.54) | 0.152 |
| High physical activity | Reference | | Reference | |
| **History of Miscarriage** | | | | |
| Yes | 0.95 (0.33 - 2.73) | 0.929 | 1.36 (1.05 - 1.77) | 0.019 |
| No | Reference | | Reference | |
| **Neonatal birth weight (grams)** | | | | |
| Quartile 1: <2900 | Reference | | Reference | |
| Quartile 2: 2900 to <3100 | 1.19 (0.51 - 2.76) | 0.676 | 0.94 (0.67 - 1.32) | 0.739 |
| Quartile 3: 3100 to <3400 | 1.30 (0.58 - 2.91) | 0.506 | 1.18 (0.88 - 1.58) | 0.262 |
| Quartile 4: ≥3400 | 1.50 (0.56 - 4.03) | 0.411 | 1.53 (1.14 - 2.06) | 0.004 |
| **Sex of infants** | | | | |
| Female | 1.18 (0.63 - 2.21) | 0.599 | 0.97 (0.79 - 1.18) | 0.791 |
| Male | Reference | | Reference | |

aOR: Adjusted Odds Ratio; CI: Confidence Interval.

birth mothers underwent episiotomy, compared to 61% of second and subsequent births mothers. This also explains that after stratifying by parity, there were no factors associated with episiotomy among first-time births. The factors associated with episiotomy in the overall analysis were primarily driven by associations observed in second and subsequent births as almost all first-time birth women underwent episiotomy, limiting the variability in factors among this group.

Maternal age was found inversely associated with episiotomy among multiparous women in our study, which is consistent with some studies, although findings have been mixed in the literature [20]. We also observed an association between higher neonatal birth weights and episiotomy, in line with findings from other studies [20–22]. However, this relationship was not observed among Vietnamese-born women in Australia [18], possibly due to the dichotomization of birth weight in that study (≥3800 grams vs. <3800 grams), which may have obscured the association. In our study, the association was observed only between the fourth and the first quartiles of birth weight.

Previous research suggests that higher levels of physical activity during pregnancy may shorten labor duration, potentially reducing the need for interventions such as episiotomy [23]. In our study, we also found that mothers in low-physical-activity occupations were more likely to undergo the procedure. This supports the idea that physical activity during pregnancy may contribute to reduce episiotomy rates. Additionally, a history of miscarriage was associated with an increased likelihood of episiotomy in our study. Previous research has shown that mothers with obstetric complications during their current pregnancy are more likely to undergo episiotomy compared to those without complications [20,21]. For obstetricians and midwives, episiotomy may be viewed as a preventive measure to reduce adverse outcomes in high-risk mothers [24]. Therefore, a history of miscarriage could increase the likelihood of episiotomy, as women with such history might be considered as high-risk in our study. Further research is needed to explore strategies for reducing episiotomy rates among mothers with a history of miscarriage, while minimizing the risk of adverse outcomes.

Strengths of our study include access to delivery data from hospital birth registers at the largest maternity hospitals in Central Vietnam, where restrictive episiotomy guidelines were first introduced in 2014. This allowed us to describe episiotomy practices in Central Vietnam for the first time and to evaluate the initial impact of the guidelines on episiotomy use. Our study is also the first to identify factors associated with episiotomy in Vietnam.

However, there are several limitations to this study. First, we lacked information on the duration of the second stage of labor, episiotomy-related complications, and perineal tear rates among women who did not undergo episiotomy. This limitation may restrict our ability to assess the full range of potential risk factors for episiotomy and limit the interpretation of episiotomy use and its relationship with maternal birth outcomes. Second, the absence of data allowing stratification of episiotomy rates by maternity care providers may limit insights into differences in clinical practice patterns. Additionally, because the study was conducted in a single hospital, generalizability of the findings may be limited. The data used in this study were collected in 2015–2016, only one year following restrictive policy of episiotomy, which may not be a sufficient time to assess the effect of the policy. Moreover, our data may not reflect current episiotomy practices. Nevertheless, our results provide valuable insights into the high rates of episiotomy and the factors associated with episiotomy practices in Vietnam. Together with findings from Hung Vuong hospital in southern Vietnam suggesting that perineal tears without episiotomy are predominantly mild if women received perineal protection [25], our findings lay the groundwork for future comprehensive research focusing on the complications and outcomes of episiotomy among Vietnamese mothers to inform strategies aimed at reducing the high rate of episiotomy in Vietnam.

## Conclusion

Our study highlights an extremely high episiotomy rate among primiparous women in Central Vietnam, one year after the implementation of restrictive guidelines, despite other neonatal and maternal factors. For multiparous women, factors such as neonatal birth weight, maternal age, occupation, and miscarriage history were associated with episiotomy. Further comprehensive studies and targeted interventions are needed to reduce episiotomy rates, particularly among first-time mothers in Vietnam.

## Acknowledgments

The authors acknowledge partial support from USAID Partnership for Higher Education Reform project for work on this manuscript.

## Author contributions

**Conceptualization:** Hoang Thi Nam Giang, Thanh-Huyen T. Vu.

**Data curation:** Hoang Thi Nam Giang, Do Thi Thuy Duy.

**Formal analysis:** Hoang Thi Nam Giang.

**Methodology:** Thanh-Huyen T. Vu.

**Validation:** Thanh-Huyen T. Vu.

**Writing – original draft:** Hoang Thi Nam Giang.

**Writing – review & editing:** Hoang Thi Nam Giang, Do Thi Thuy Duy, Thanh-Huyen T. Vu.

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
