## [Editor Report · Decision Letter 0]

6 Mar 2025

PONE-D-25-10826Episiotomy Practices and Associated Factors in Central VietnamPLOS ONE

Dear Dr. Giang,

Thank you for submitting your manuscript to PLOS ONE. After careful consideration, we feel that it has merit but does not fully meet PLOS ONE’s publication criteria as it currently stands. Therefore, we invite you to submit a revised version of the manuscript that addresses the points raised during the review process.

We look forward to receiving your revised manuscript.

Kind regards,

Chenchit Pichailuck, Associate Professor

Academic Editor

PLOS ONE

Journal Requirements:

2. Please note that your Data Availability Statement is currently missing [the repository name and/or the DOI/accession number of each dataset OR a direct link to access each database]. If your manuscript is accepted for publication, you will be asked to provide these details on a very short timeline. We therefore suggest that you provide this information now, though we will not hold up the peer review process if you are unable.

Additional Editor Comments:

-The rationale of the study is not clearly stated. The national policy of Veitnam to reduce episiotomy practice has been launched one year prior to the data collection. The readers may be curious to know how things have been done. Moreover, why the auhtors choose 'one year following the policy' to represent the situation when there could some consequential effects from prior-to-policy practice.

-The conclusion of the abstract shows that 'one year after the implementation of..' comes without background information.

-Page 4 Line 51: I doubt with the advantages of episiotomy over spontaneous tear in that 'healing is better', 'preserve relaxation of the pelvic floor m.', I think that it should be the opposite direction
---

## [Author Response · Author response to Decision Letter 1]

18 Apr 2025

Dr. Chenchit Pichailuck, Associate Professor

Academic Editor

PLOS ONE

Re: Episiotomy Practices and Associated Factors in Central Vietnam

Date: April 18, 2025,

Dear Dr. Pichailuck,

Thank you very much for the opportunity to revise this manuscript. We have addressed the comments in the point-by-point response below.

Both tracked-changes version and a clean version of the revised manuscript are included.

We sincerely appreciate your valuable feedback and suggestions, which have significantly improved the quality of our work.

We hope that our revision has met PLOS ONE’s publication criteria and look forward to your response!

Yours sincerely,

Drs. Hoang Thi Nam Giang & Thanh-Huyen T. Vu

Editor Comments

Comment: The rationale of the study is not clearly stated. The national policy of Veitnam to reduce episiotomy practice has been launched one year prior to the data collection. The readers may be curious to know how things have been done. Moreover, why the auhtors choose 'one year following the policy' to represent the situation when there could some consequential effects from prior-to-policy practice.

Response: We have revised the introduction to clarify the rationale behind our study. Currently, there is limited data on episiotomy practices in Vietnam and no data on the factors associated with this procedure. While a few studies have reported episiotomy prevalence in Northern and Southern Vietnam, none have been done in Central Vietnam; and none have focused on identifying the contributing factors. Our study aims to help fill this gap and we hope that our results can provide a foundation for future research on the complications and outcomes of episiotomy among Vietnamese mothers.

The study is based on previously collected data, and unfortunately, we do not have data for subsequent years. We discussed this issue as one of our limitations. We also removed the mention of the national policy to avoid misleading readers about its impact, as this could not be adequately addressed in this current study. The revision as follows:

Introduction: Page 4 (line 65-67) and page 5 (line 68-82):

“In Vietnam, the rate of episiotomy use and its associated factors are rarely reported despite the 2014 policy change that no longer recommends routine episiotomy(8). There was a survey conducted between 2012-13 in a maternity hospital in southern Vietnam to assess clinicians’ practices regarding episiotomy (9). In this study, among nulliparous women, the majority of obstetricians (82.6%) and midwives (98.7%) reported using episiotomies more than 90% of the time. Among multiparous women, a quarter of obstetricians reported using episiotomies under 60% of the time, while this rate for midwives was 3.8%. In 2020, the rate of episiotomy was reported in a study conducting that focusing on breastfeeding in both southern and northern Vietnam, where 83.3% of mothers self-reported having undergone a vaginal birth with an episiotomy (10).

To our knowledge, no comprehensive study has been conducted on episiotomy practices and their associated factors in Vietnam, particularly in Central Vietnam. Therefore, this study aimed to describe the prevalence of episiotomy among women who had vaginal births of single, full-term infants, and to identify the maternal and infant factors associated with this practice.”

Page 15 (line 235-236)

“The data used in this study were collected in 2015-2016 and therefore may not reflect current episiotomy practices”.

Comment: The conclusion of the abstract shows that 'one year after the implementation of..' comes without background information.

Response: We have removed the reference to policy information and added further details on modifiable factors associated with episiotomy practice in the conclusion, as follows:

Page 3 (line 34-38)

“This study highlights a very high episiotomy rate in Central Vietnam, especially among primiparous women. Several modifiable maternal and neonatal factors are linked to episiotomy use in multiparous women. Comprehensive research and targeted interventions are needed to reduce episiotomy rates, particularly among first-time mothers in Vietnam.”

Comment: Page 4 Line 51: I doubt with the advantages of episiotomy over spontaneous tear in that 'healing is better', 'preserve relaxation of the pelvic floor m.', I think that it should be the opposite direction.

Response: We have removed these references.

---

## [Editor Report · Decision Letter 1]

13 May 2025

PONE-D-25-10826R1Episiotomy Practices and Associated Factors in Central VietnamPLOS ONE

Dear Dr. Giang,

Thank you for submitting your manuscript to PLOS ONE. After careful consideration, we feel that it has merit but does not fully meet PLOS ONE’s publication criteria as it currently stands. Therefore, we invite you to submit a revised version of the manuscript that addresses the points raised during the review process.

We look forward to receiving your revised manuscript.

Kind regards,

Chenchit Pichailuck, Associate Professor

Academic Editor

PLOS ONE

Additional Editor Comments:

-Episiotomy has been commonly practiced in both Thailand and Veitnam because the procedure is included in the Medical curriculum whereas the global trend goes to the opposite direction. Many countries in this region such as Malaysia and Singapore minimize the procedure by increasing the perineal preparation such as massaging, breathing exercise during antenatal care, which takes extra time and effort. The high prevalence of episiotomy in Veitnam can be lessened by the strong Government policy which was mentioned in the original version of manuscript. And, I think it would be very interesting to learn from how the Veitnamese Government deal with the problem. However, the authors decide to remove this part resulting in lower added value of the study.

-According to the study, episiotomy is commonly practicec in all deliveries. I am not sure if around 30% of multiparous women who did not undergo episiotomy had too short second stage to receive episiotomy or not. Second, third stage of labour; post-partum hemorrhage and instrumental deliveries should be taken into consideration.

-Conceptual framework of the study should be re-considered.

---

## [Author Response · Author response to Decision Letter 2]

11 Nov 2025

Dr. Chenchit Pichailuck, Associate Professor

Academic Editor

PLOS ONE

Re: Episiotomy Practices and Associated Factors in Central Vietnam

Date: November 11, 2025,

Dear Dr. Pichailuck,

Thank you very much for the opportunity to revise this manuscript. We have addressed the comments in the point-by-point response below.

Both tracked-changes version and a clean version of the revised manuscript are included.

We sincerely appreciate your valuable feedback and suggestions, which have significantly improved the quality of our work.

We hope that our revision has met PLOS ONE’s publication criteria and look forward to your response!

Yours sincerely,

Drs. Hoang Thi Nam Giang & Thanh-Huyen T. Vu

Editor Comments

Comment: Episiotomy has been commonly practiced in both Thailand and Veitnam because the procedure is included in the Medical curriculum whereas the global trend goes to the opposite direction. Many countries in this region such as Malaysia and Singapore minimize the procedure by increasing the perineal preparation such as massaging, breathing exercise during antenatal care, which takes extra time and effort. The high prevalence of episiotomy in Veitnam can be lessened by the strong Government policy which was mentioned in the original version of manuscript. And, I think it would be very interesting to learn from how the Veitnamese Government deal with the problem. However, the authors decide to remove this part resulting in lower added value of the study.

Response: We are sorry for misunderstanding your comments. We have now added information on the government policy in the abstract, introduction and the discussion.

The revision as follows:

Abstract: Page 3 (line 35-42)

“This study highlights a very high episiotomy rate in Central Vietnam, especially among primiparous women despite recommendations from the World Health Organization and the Viet Nam Ministry of Health. Several modifiable maternal and neonatal factors are linked to episiotomy use in multiparous women. While one year may not be sufficient to fully assess the restrictive policy's effect, the results of our study also suggest that the policy has not had a significant impact on reducing the episiotomy rate after one year. This indicates that, in addition to the policy, further support and interventions are needed to reduce episiotomy rate”

Introduction: Page 4 (line 64-66) and page 5 (line 67-86)

“In Vietnam, the rate of episiotomy use and its associated factors are rarely reported despite the 2014 policy change that no longer recommends routine episiotomy (8). Prior to 2014, only one survey conducted between 2012-13 in a maternity hospital in southern Vietnam to assess clinicians’ practices regarding episiotomy (9). In this study, among nulliparous women, the majority of obstetricians (82.6%) and midwives (98.7%) reported using episiotomies more than 90% of the time. Among multiparous women, a quarter of obstetricians reported using episiotomies under 60% of the time, while this rate for midwives was 3.8%. In 2014, the Vietnam Ministry of Health approved a technical guide on essential care for mothers and newborns during and immediately after birth (EENC), the practice of routine episiotomy is no longer recommended (8). Since then, there is lacking data to evaluate the effect of this restrictive recommendation. In 2020, the rate of episiotomy was reported in a study conducting that focusing on breastfeeding in both southern and northern Vietnam, where 83.3% of mothers self-reported having undergone a vaginal birth with an episiotomy (10).”

“To our knowledge, no comprehensive study has been conducted on episiotomy practices and their associated factors in Vietnam, particularly in Central Vietnam. Moreover, although routine episiotomy has not been recommended since the implementation of EENC guideline in 2014, and EENC was introduced in Da Nang Hospital for Women and Children the same year, data on episiotomy practice during the early stage of this transition remain limited. Therefore, this study aimed to describe the prevalence of episiotomy among women who had vaginal births of single, full-term infants after 1-year period of the implementation of restrictive episiotomy guidelines in Vietnam, and to identify the maternal and infant factors associated with this practice”.

Discussion: Page 12 (line 176-177), page 12, 13 (line 182-185), page 15 (line 240-241), and page 16 (line 250-253)

“This study included 3,471 singleton, full-term vaginal births in Central Vietnam during 2015-2016, one year following the implementation of restrictive episiotomy guidelines launched in the country”

“The high episiotomy rate in our study aligns with the finding from existing literature in other regions of the country. A survey conducted in 2012-2013 among obstetricians and midwives in southern Vietnam - prior to the implementation of restrictive episiotomy guidelines by the Vietnam Ministry of Health in 2014 - reported that 82.6% of obstetricians and 98.7% midwives performed episiotomies more than 90% of the time for first-time births (9)”

“Strengths of our study include access to delivery data from hospital birth registers at the largest maternity hospitals in Central Vietnam, where restrictive episiotomy guidelines were first introduced in 2014”

“The data used in this study were collected in 2015-2016, only one year following restrictive policy of episiotomy, which may not be a sufficient time to assess the effect of the policy. Also, our data may not reflect current episiotomy practices”

Comment: According to the study, episiotomy is commonly practicec in all deliveries. I am not sure if around 30% of multiparous women who did not undergo episiotomy had too short second stage to receive episiotomy or not. Second, third stage of labour; post-partum hemorrhage and instrumental deliveries should be taken into consideration.

Response: The study is based on previously collected data, and unfortunately we do not have data on the duration of second and third stage of labour and other information to add in the results.

Comment: Conceptual framework of the study should be re-considered.

Response: Thank you for your suggestion. We have added goverment policy as an important part in our manuscript.

---

## [Editor Report · Decision Letter 2]

1 Jan 2026

PONE-D-25-10826R2Episiotomy Practices and Associated Factors in Central VietnamPLOS One

Dear Dr. Giang,

Thank you for submitting your manuscript to PLOS ONE. After careful consideration, we feel that it has merit but does not fully meet PLOS ONE’s publication criteria as it currently stands. Therefore, we invite you to submit a revised version of the manuscript that addresses the points raised during the review process.

We look forward to receiving your revised manuscript.

Kind regards,

Berhanu Elfu Feleke, Ph.D.

Academic Editor

PLOS One

Journal Requirements:

Additional Editor Comments:

• Please report interval estimates (e.g., 95% confidence intervals) consistently for all key effect measures to allow readers to assess the precision of the estimates.

• Please clarify who performed the episiotomies (e.g., midwives versus obstetricians/gynaecologists). If these data are available, consider stratifying episiotomy rates by provider type, as this may provide important insights into clinical practice patterns.

• If available, please report the perineal tear rates among women who did not undergo episiotomy (n = 701). This information would strengthen the interpretation of episiotomy use and its relationship to maternal birth outcomes.

---

## [Author Response · Author response to Decision Letter 3]

3 Feb 2026

Dr. Berhanu Elfu Feleke, Ph.D

Academic Editor

PLOS One

Re: Episiotomy Practices and Associated Factors in Central Vietnam

Date: 2 January 2026

Dear Dr. Berhanu Elfu Feleke,

Thank you very much for the opportunity to revise this manuscript. We have addressed the comments in the point-by-point response below.

Both tracked-changes version and a clean version of the revised manuscript are included.

We sincerely appreciate your valuable feedback and suggestions, which have significantly improved the quality of our work.

We hope that our revision has met PLOS One’s publication criteria and look forward to your response!

Yours sincerely,

Drs. Hoang Thi Nam Giang & Thanh-Huyen T. Vu

Editor Comments

Comment: Please report interval estimates (e.g., 95% confidence intervals) consistently for all key effect measures to allow readers to assess the precision of the estimates.

Response: We reported 95% confidence intervals for all key effect measures.

The revisions are as follows:

Abstract: Page 2 (line 23-27)

“Among 3,471 eligible singleton births, 2,770 mothers (79.8%, 95% CI: 78.4% - 81.1%) underwent an episiotomy. The episiotomy rate was significantly higher in first-time births (97.7%, 95% CI: 96.8% - 98.3%) compared to second or subsequent births (61.5%, 95% CI: 59.2% - 63.8%), p<0.001”

Result: Page 8 (line 145-147)

“Among the 3,471 births, the mean maternal age was 28.4, 2,770 (79.8%, 95% CI: 78.4% - 81.1%) involved an episiotomy. The episiotomy rate was 97.7% for first-time births (95% CI: 96.8% - 98.3%) and 61.5% for second or subsequent births (95% CI: 59.2% - 63.8%)”

Table 1: Page 9-11 (All measures in Table 1 were added 95%CI)

Table 1. Maternal and Neonatal Characteristics, Overall and by Parity, n=3,471

Comment: Please clarify who performed the episiotomies (e.g., midwives versus obstetricians/gynaecologists). If these data are available, consider stratifying episiotomy rates by provider type, as this may provide important insights into clinical practice patterns.

Response: We do not have data that allows stratification of episiotomy rates by provider type, and we acknowledge this is a limitation of our study. We have added further details to the Limitation section accordingly.

The revision is as follows:

Discussion: Page 16 (line 248-255)

“However, there are several limitations to this study. First, we lacked information on the duration of the second stage of labor, episiotomy-related complications, and perineal tear rates among women who did not undergo episiotomy. This limitation may restrict our ability to assess the full range of potential risk factors for episiotomy and limit the interpretation of episiotomy use and its relationship with maternal birth outcomes. Second, the absence of data allowing stratification of episiotomy rates by maternity care providers may limit insights into differences in clinical practice patterns”

Comment: If available, please report the perineal tear rates among women who did not undergo episiotomy (n = 701). This information would strengthen the interpretation of episiotomy use and its relationship to maternal birth outcomes.

Response: As noted in the Limitation section (page 16, line 248-255) and in our previous response, we do not have data on perineal tear rates among women who did not undergo episiotomy in our study population. We agree that the absence of this information limits the interpretation of episiotomy use and its relationship with maternal birth outcomes. However, data from Hung Vuong hospital in southern Vietnam, based on a review of 367 post-partum medical records of women who received perineal protection, showed that 77.9% experienced perineal tear; among these, 94% were first-degree tears, and no third- or fourth-degree tears were reported. These finding suggest that perineal tears without episiotomy are predominantly mild and support the need for further comprehensive research to inform strategies aimed at reducing the high rate of episiotomy in Vietnam. We have included these comments in our discussions (page 16 lines 260-262 and page 17 lines 263-265).

“Together with findings from Hung Vuong hospital in southern Vietnam suggesting that perineal tears without episiotomy are predominantly mild if women received perineal protection (25), our findings lay the groundwork for future comprehensive research focusing on the complications and outcomes of episiotomy among Vietnamese mothers to inform strategies aimed at reducing the high rate of episiotomy in Vietnam”

---

## [Editor Report · Decision Letter 3]

8 Feb 2026

Episiotomy Practices and Associated Factors in Central Vietnam

PONE-D-25-10826R3

Dear Dr. Giang,

We’re pleased to inform you that your manuscript has been judged scientifically suitable for publication and will be formally accepted for publication once it meets all outstanding technical requirements.

Kind regards,

Berhanu Elfu Feleke, Ph.D.

Academic Editor

PLOS One
---

## [Editor Report · Acceptance letter]

PONE-D-25-10826R3

PLOS One

Dear Dr. Giang,

I'm pleased to inform you that your manuscript has been deemed suitable for publication in PLOS One. Congratulations! Your manuscript is now being handed over to our production team.

Kind regards,

on behalf of

Dr Berhanu Elfu Feleke

Academic Editor

PLOS One